# An exploration of proactive health oriented symptom patterns in patients undergoing percutaneous coronary intervention with stent implantation: A mixed-methods study protocol

Qi Wang[1], Chaoyue Xu[1], Zhiqing He[1], Ping Zou[2], Jing Yang[3], Yanjin Huang[1]*

**1** School of Nursing, Hengyang Medical School, University of South China, Hengyang, China, **2** Nipissing University, Toronto, Ontario, Canada, **3** The Second Affiliated Hospital of University of South China, Hengyang, China

* 2019000050@usc.edu.cn

**Data Availability Statement:** No datasets were generated or analyzed during the current study. All

## Abstract

### Background

Coronary Heart Disease (CHD) is one of the most prevalent chronic diseases worldwide. Currently, percutaneous coronary intervention (PCI) with stent implantation is the main clinical treatment for CHD, and patients can achieve better outcomes after stenting. However, adverse cardiovascular events continue to recur, ultimately failing to yield good results. Several symptoms exist after stenting and are associated with health outcomes. Little is known about the symptom patterns of patients during the different postoperative periods. Therefore, this study aims to explore the dynamics of symptoms and clarify the experiences of post-stenting in patients during different periods, which may help the delivery of more specific patient management and improve survival outcomes in the future.

### Methods

A mixed method (quantitative/qualitative) design will be adopted. Longitudinal research, including surveys regarding three different periods, will be sued to describe the symptom patterns of patients undergoing PCI with stent implantation, clarifying their focused symptom problems during different time periods or in populations with different features. Qualitative individual interviews aim to understand the feelings, experiences, opinions, and health conditions of patients post-stenting, which can explain and supplement quantitative data. Quantitative data will be analyzed using descriptive statistics, latent class analysis (LCA), and latent translation analysis (LTA). Qualitative data will be analyzed using content analysis.

### Discussion

This study is the first study to explore the symptom patterns and experiences of patients in various domains after stent implantation using a novel design including quantitative and

relevant data from this study will be made available upon study completion.

**Funding:** This study was supported by the Health Commission of Hunan Province (Jing Yang, NO. B202314057822). The funders did not and will not have a role in study design, data collection and analysis, decision to publish, or preparation of the manuscript.

**Competing interests:** The authors have declared that no competing interests exist.

qualitative methods, which will help the delivery of more specific patient management, reduce the recurrence of adverse cardiovascular events, and improve survival outcomes in the future. It is also meaningful to use PROMIS profile-57 to help patients to proactively focus on their health problems, promote health literacy, and incorporate active patient participation into health management, which is a successful transition from passive medical treatment to active management.

## Introduction

Coronary atherosclerotic heart disease, also known as coronary heart disease (CHD), is an escalating public global health problem that is a major cause of death in humans [1–3]. According to the latest data from the American College of Cardiology, the total disability-adjusted life years (DALYs) was approximately 182 million, and 197 million people suffered from coronary heart disease, eventually resulting in 9.14 million deaths worldwide in 2019 [4]. The mortality and incidence of CHD have been increasing even in younger people and the disease may become a leading cause of death in China in the near future [5].

Following significant advances in techniques such as stent implantation, the preferred treatment for CHD is percutaneous coronary intervention (PCI) [6]. Coronary artery stent implantation (CASI), the most common type of PCI, has become the most prevalent revascularization strategy for patients with CHD. It can restore coronary blood flow rapidly and improve blood supply, with less trauma and effective improvement in patient prognosis [7–9]. Although patients can obtain beneficial results after stent implantation, they cannot reverse or slow down the pathological process of coronary atherosclerosis [10]. Adverse cardiovascular events, including second restenosis and thrombosis, occur and have become serious clinical problems [11]. A systematic review and meta-analysis showed that the rate of in-stent restenosis in patients undergoing CASI is still higher than 10% [12]. These patients still contend with rehospitalization risks and increased medical expenses [13]. Promoting the recovery process and achieving an excellent prognosis is a setback that needs to be solved.

Usually, patients who have undergone PCI with stent implantation develop postoperative psychological, physical, and social health complications, which are related to their rehabilitation process [14]. Several studies have shown that patients with stent implantation are more likely to experience more negative emotions than patients treated without stenting [15, 16]. In all interviews in the qualitative study, patients after coronary stenting expressed some degree of anxiety, mainly concerning the procedure and uncertainty about the future [17]. Shen [18] emphasized that the surgery is invasive, and patients may experience psychological problems such as depression due to economic problems caused by stenting, long-term postoperative medication, and fear of surgical complications. Indeed, these negative psychological situations can reduce patient compliance with treatment and significantly decrease clinical benefit [19]. Pain is a common post-operative symptom. Persistence or recurrence of angina after stent implantation may affect approximately 20–40% of patients during short- to medium-term follow-up [20]. It is well known that aerobic exercise is essential for rehabilitating patients with cardiovascular diseases. However, avoidance of exercise is common in patients undergoing PCI with stent implantation, leading to impaired exercise tolerance and decreased physical activity [21]. Several scholars have concluded that physical defects are strongly associated with health outcomes in patients post-stenting, specifically in terms of delaying the process of atherosclerosis, improving long-term survival, and reducing the recurrence of cardiovascular

events [22–24]. Finally, regarding social dimension symptoms, patients may refuse to return to their normal work because of fear of relapse, resulting in a lack of social function. As shown in one study, the Social Disability Screening Schedule (SDSS) scores were mostly high in patients post-stenting, which implies the presence of severe social dysfunction [25]. If all symptoms cannot be promptly identified and managed, recurrence of cardiac events may also occur, eventually causing serious adverse effects on patient prognosis [26]. Hence, understanding the occurrence of symptoms in patients post-stenting is necessary.

Recently, studies have gradually begun to focus on symptom management of patients after stent implantation, with the aim of reducing postoperative cardiac events. Researchers have used cardiac rehabilitation to manage patients post-stenting to reduce the incidence of adverse events and improve negative symptoms and quality of life [27, 28]. However, management programs, including universal general care, medication care, and mental nursing, tend to follow a uniform model for the whole population, meaning that the same set of management protocols are applied to patients with different social backgrounds, disease processes, health situations, and social support. Such protocols lack exploration of individual symptom patterns, making it difficult to meet the postoperative health needs of each patient. Therefore, the main barriers to postoperative management lie in exploring individual symptom patterns, which are reflected in two aspects. Notably, the symptoms of the disease vary by population. An integrative review of factors influencing physical activity after cardiac surgery concluded that the elderly exhibited more serious physical activity problems [29]; this may be explained by atrophy of skeletal muscles, reduced range of joints, and decreased physical function with age. However, social function issues are more prominent in young and middle-aged populations. As the breadwinners and spiritual support of their families, patients need to consider how to maintain their social and family role obligations [30]. Patients with combined underlying diseases, such as hypertension and diabetes, are more likely to be depressed and anxious after stenting, which may be related to the fact that hypertension and diabetes are typically chronic diseases, with psychological problems being more pronounced [31, 32]. In addition, owing to the psychological and physiological peculiarities of female patients, they are at a high risk of depressive symptoms [33]. Patients with emergency stenting tend to show more psychological disturbances due to the unpredictability of unexpected situations [17]. The population with secondary stent implantation may be more likely to have a self-perceived burden due to the high number of stent placements [34]. However, the symptoms are not invariable in the same patient. Post-stenting symptoms are a dynamic process that interact with each other, change over time, or are affected by other factors. Due to surgical treatment, anterior heart pain in patients post-stenting may gradually decrease. Simultaneously, early postoperative pain may cause negative emotions and fear of activity, which may lead to limitations in physical activity [35]. A prospective cohort study found that some patients experienced deteriorated mental health after discharge, and psychological symptoms were more negative at 1 month after discharge compared with other periods resulting from uncertainties during illness and patients feeling annoyed by troublesome tasks after discharge [36]. Overall, the comprehension of symptom patterns requires further exploration to better meet the management needs of different patients at different times, improve quality of survival, and avoid adverse events.

This protocol is based on the concept of proactive health, which focuses on integrated physiological, psychological, and social functions and emphasizes proactive participation in health management. By shifting from reactive disease care to proactive health management, patients can be empowered to understand and maintain their health. Patient-reported outcomes (PRO) will be introduced in our study, which accesses the health status of patients that is directly derived from the active reporting by patients, with no interpretation by medical providers or others [37]. This allows the management process to incorporate the voices of patients,

such that the study can help patients to understand their health problems, contributing to their health skills and behavior. Addressing the setback in post-stenting management to reduce adverse cardiac events, our study aims to (1) explore post-stenting symptom patterns at different time periods or in populations with unique characteristics; and (2) clarify the symptom experience of patients undergoing PCI with stent implantation, which this is critical to understanding the individual needs of patients.

## Methods

### Study design overview

This study will apply a mixed-methods design. The longitudinal design, which includes surveys regarding three different periods (post-stenting hospitalization, 3 months after discharge, 6 months after discharge) based on the cardiac rehabilitation stages [38], will be used in the quantitative phase to explore post-stenting symptom patterns of patients and determine the different symptom characteristics during different periods or in different populations using the Patient-Reported Outcomes Measurement Information System (PROMIS), which is able to comprehensively assess symptoms in the physical, psychological, and social health dimensions [39]. The descriptive qualitative design involves several semi-structured interviews aiming to learn about symptom experience, provide an explanation for and supplement quantitative data, or provide an in-depth understanding of the significant, unexpected, confused, or unexplained results that may occur during the quantitative phase, such as patients who reported extreme symptom scores. The quantitative phase is the dominant area of research, with the qualitative phase playing an explanatory and supplementary role. When two kinds of data are collected, the results will be integrated to provide insights into the symptom patterns of different patients post-stenting at different postoperative periods.

### Quantitative phase (longitudinal study)

**Participants.** Based on the guidelines for cardiovascular rehabilitation and secondary prevention in China [39], this section will longitudinally track the symptoms of post-stenting patients in rehabilitation phase I (post-stenting hospitalization, T1), rehabilitation phase II (3 months after discharge, T2), and rehabilitation phase III (6 months after discharge, T3), for a total of three batches of surveys. Those that are eligible for this study recruitment are as follows: patients admitted with a confirmed diagnosis of coronary heart disease who have undergone successful PCI with stent implantation, patients who reside in a community within the urban area of Hengyang City, Hunan Province, China, disclose no reading comprehension difficulties, and voluntarily enroll in our study. Meanwhile, patients with mental abnormalities of any type and malignant disease according to clinical criteria, and those who have vague consciousness and cannot cooperate with the researcher will be excluded from participation in the study. During the following T2 and T3 surveys, patients who experience further major adverse cardiovascular events, die, refuse follow-up, or choose to withdraw from the study will also be excluded.

**Sample and study setting.** There are many tertiary and teaching hospitals in Hengyang City, with high-quality service resources. These hospitals have established bases for medical research activities with community service units and undertake public health work for the resident population, providing a source of research subjects and community medical resources for conducting this study. We will select hospitals in Hengyang City using the cluster sampling method. Eventually, three hospitals will be randomly selected. All post-stenting patients in the selected three hospitals from March 1, 2023, to May 31, 2023, who meet the inclusion and

exclusion standards, will be included and excluded, respectively, in this T1 survey, and we will continue to perform the T2 and T3 surveys at 3 and 6 months after discharge.

We aim to explore the symptom patterns of patients with CHD after stent implantation using Latent Transition Analysis (LTA) which is based on Latent Class Analysis (LCA) and enables the researcher to model dynamic changes. The estimation of sample size for Latent Class Model (LCM) is currently being explored. Most studies suggest 500 as the worthy goal in practice [40]. Therefore, we will recruit a total of 600 eligible patients who meet the inclusion criteria from three tertiary hospitals in Hengyang City with an expected dropout rate of 20%. Each hospital will enroll 200 patients through convenience sampling.

**Data collection.** Before collecting the data, the study participants will be informed of the purpose, implementation steps, and ethical principles of the study, and their right to informed consent will be fully ensured. The research instruments will be distributed by uniformly trained staff. The study participants will be required to complete the questionnaires and submit these immediately on completion, and those who may be unable to complete the questionnaire themselves will be able to dictate their responses to a family member who will record them. During this phase, data will be collected using two instruments. A general questionnaire will be designed by the research team to collect the research general personal and health characteristics of the participants. The questions will include general information (such as age, sex, residence, marital status, educational level, occupation, economic income, medical insurance type, smoking, drinking, height, and weight) and details about the disease (such as duration of CHD, family history of CHD, cardiac function grading, comorbid chronic diseases, postoperative time, and the number of stents implanted). We will also use the following measurements to collect statistics on symptom scores.

The Patient-reported Outcomes Measurement Information System (PROMIS) was initiated by National Institutes of Health. Evon [41] conducted a study among patients diagnosed with chronic hepatitis C virus to evaluate the reliability and validity of 10 PROMIS short forms, including fatigue, depression, anxiety and sleep disturbance. The PROMIS measures exhibited favorable reliability (Cronbach's α coefficient ≥0.87) and demonstrated moderate to strong associations with theoretically-similar items from specific scales assessing symptoms in patients with HCV (0.39–0.77). Additionally, all the data provided strong support for the structural validity of PROMIS measures using item-response-theory models. Krohe [42] performed a cognitive debriefing interview to evaluate the content validity of PROMIS short term regarding physical function, and showed that more than 90% of individuals demonstrated understanding of each item in the PROMIS. The selection of informative items for the PROMIS profile-57 consisting of seven short terms was based on a comprehensive literature review, item response theory analysis, and expert review of psychometric evaluation results from the PROMIS datasets. The original English version of the PROMIS profile-57 has excellent brevity, breadth, and robust psychometric properties. Furthermore, it has been successfully translated into over 40 cross-cultural adapted versions, all of which have shown satisfactory psychometric properties in various clinical conditions [43]. It contains seven fixed PROMIS core domains, including physical function, anxiety, depression, fatigue, sleep disturbance, ability to participate in social roles and activities, and pain interference. Each of these domains has eight items with a score of 8–40 and includes a pain intensity item measured on a scale of 0–10 [44]. All domains are scored on a 5-point Likert scale, and there are differences between the descriptions of the five options for each item, except for the pain intensity item [45]. Each short form consisting of eight items in Profile-57 was scored separately, resulting in scores for a total of seven domains. In addition, the scores were converted to standardized T-scores with a mean of 50 and a standard deviation of 10 [46]. Higher T-scores reflect higher levels of the respective measured concepts. Among them, higher scores for positive scoring domains indicate better

function or fewer symptoms. In contrast, higher scores for negative scoring domains indicate impaired function or more pronounced symptoms. For example, a T-score of 70 in the short term about social roles and activities indicates that the ability of a participant to perform social roles and participate in social activities is higher than the average value for the general population, and the social function of the patient is good. Conversely, a T-score of 70 in short-term depression indicates that the depression in the patient is higher than the average value for the general population, and the patient has a higher level of depression [47, 48].

Shah indicated that self-reporting using PROMIS is recommended for patients with cardiovascular disease who have a high level of self-care awareness. It can help clinical staff to better understand the needs of patients and enhance their engagement with the disease, which has positive implications for patient health [49]. Prof. Yuan's team developed a simplified Chinese version of the PROMIS profile-57 in accordance with the translation guidelines and conducted a psychometric evaluation in patients with breast cancer demonstrating acceptable convergent validity, discriminant validity, criterion validity, and reliability. In their study, the correlations between each item with its domains were all greater than 0.40 in multi-trait scaling analysis, the correlations between PROMIS profile-57 item scores with the corresponding domains coefficients in the Functional Assessment of Cancer Therapy-Breast ranged from 0.32–0.56 using the Pearson correlation test, and Cronbach's α coefficients ranged from 0.85 to 0.95 [50]. We will use the Chinese version of the PROMIS profile-57 to measure the patients with post-stenting symptom scores in these seven domains and to explore the different symptom categories among patients in these three different periods (T1: post-stenting hospitalization; T2: 3 months after discharge; T3: 6 months after discharge) or in patients with different characteristics.

**Analysis of quantitative data.**   All data will be recorded and described using IBM SPSS version 26.0 (IBMCorp., Armonk, N.Y., USA). Descriptive statistics will be calculated for general personal and health characteristics in the sample, in which continuous variables will be analyzed by means and standard deviations, and categorical variables will be described in frequency and percentages. We will transform all PROMIS-57 raw scores of patients into standardized T-scores based on PROMIS guidelines (http://www.healthmeasures.net). In addition, Latent Class Analysis and Latent Translation Analysis will be performed using the Mplus version 8.3 (Muthén & Muthén, Los Angeles, CA) to classify patients with post-stenting into different latent class groups and reveal the dynamic shift of symptom situation according to the translation probability of each potential category. The logistic regression will be applied to explore the influencing factors of potential symptom category conversion.

## Qualitative phase (semi-structured interview)

**Sample and study setting.**   Purposive sampling will be used to select the research participants for semi-structured interviews from each phase (T1, T2, and T3) of the quantitative phase. The final sample size of the study population will be determined by the criterion that the data are saturated and no new themes emerge [51]. As the patient is still in the hospital, we will conduct a one-on-one in-depth interview in the department conference room. For patients who will be discharged from the hospital, we will choose to meet with them in their homes or community meeting rooms. In these interview settings there will be no interference from other people.

**Data collection.**   Data will be collected through in-depth, semi-structured interviews. The interviews will be conducted by a researcher with qualitative research experience and a long history of cardiovascular disease management, and a graduate student in the field of community-based chronic disease management. Prior to conducting the interview, we will contact the

interviewee, obtain informed consent from the interviewee, and agree on a time and place for the interview. The interviews will be audio-recorded live, and the major interviewer who is responsible for communication can take notes. The secondary interviewer can carefully observe and record the expressions, tone of voice, gestures, and other non-verbal behaviors of the interviewees that can reflect psychological changes. Each interview is planned to last for 30–40 minutes.

We have developed an interview outline comprising focused open-ended questions for the semi-structured interviews: (1) What were your thoughts after you learned that you needed to undergo PCI with stent implantation? (2) What discomfort have you experienced after stenting? (3) What do you think is the most serious of these problems? (4) What is the biggest change in your life after stenting? (5) What kind of help do you want at this stage?

**Analysis of qualitative data.**   Interviews will be analyzed using content analysis, which can describe patient health status after stenting, explore their symptom trouble in different postoperative periods, and understand the symptom scores in the quantitative phase according to the subjective descriptions. We will use NVivo version 10 software to manage qualitative data analysis. First, within 24 hours of the interview, the interviewer will transcribe the recording into an electronic document verbatim. After transcription, the text will be checked for accuracy by two subject members to ensure accuracy and avoid the addition of the personal opinions and ideas of the interviewer. It is also necessary to refer to the recorded notes to supplement the non-verbal information from the interview. Second, we will open code the transcribed text material. Next, forming different categories or themes will be the critical process, in which researchers will further refine and elevate opinions based on the words used, including word frequency and relationships between words [52]. Finally, we can describe the feelings and experiences of patients post-stenting for a specific period through abstract categories or themes.

**Integration of the data.**   This is a mixed-methods research consisting of two stages. The process of replenishing or explaining each other between the qualitative and quantitative findings of a study is an important element of a mixed methods study [53].

In the quantitative study, we aim to explore symptom categories and their dynamic shift patterns in patients undergoing stent implantation through the three-stage longitudinal survey. First, latent class analysis will be conducted separately for T1, T2, and T3 to classify post-stenting patients into different homogeneous symptom categories at each postoperative stage. A series of LCA models will be performed in order to identify the optimal classification. The number of latent classes will be gradually increased until adding an additional category no longer improves the fit index of the model. Next, we will compare the evaluation criteria of all the LCA models to determine the best model, including the Akaike information criterion (AIC), Bayesian information criterion (BIC), adjusted BIC (aBIC), entropy, Lo-Mendell-Rubin likelihood-ratio test (LMR-LRT), bootstrap likelihood-ratio test (BLRT), and theoretical basis. The best model will be expected to possess lower values for the AIC, BIC, and aBIC, higher entropy, and significant LMR-LRT and BLRT results ($P<0.05$) [46]. The same modeling procedure will be applied to latent transition analysis using the longitudinal data. To ensure that the latent classes can be defined consistently across time, the restriction of measurement invariance across three different time points will be implemented. The full information maximum likelihood (FIML) was performed to handle missing values in our LTA model. This method is better than most alternative missing values methods used in longitudinal studies [54]. In LTA, a patient's latent symptom category will be allowed to change over time, and the transition probabilities to different symptom categories during different periods (T1-T2, T2-T3) will be presented, revealing the dynamic changes of individual symptoms over time. Finally, logistic regression will be used to explore factors including demographic and clinical characteristics in the general questionnaire influencing the transition of latent symptom classes at T1, T2, and T3.

The qualitative data will provide an interpretation and complement the quantitative data. In the qualitative study, the descriptions of health conditions and feelings of patients at different times can explain the symptom situation at different stages well in the quantitative results. In-depth face-to-face interviews with patients will help us to determine the specific impact of surgery on their lives and to understand the possible unexpected scores in the quantitative results. Moreover, the descriptions of the management needs of patients at different stages, corresponding to the symptom scores in the quantitative data (the more severe the symptoms, the higher the management needs), can complement what is not found in the quantitative exploration.

In conclusion, it is pertinent to integrate both quantitative and qualitative data. Both types of data will describe the symptom problems of patients during different time periods after surgery. The combination of qualitative and quantitative data will help us to better understand and supplement the symptom situation of patients at different stages or the symptom variance of patients with different characteristics and further clarify the individual management needs of patients post-stenting. This provides a basis for performing accurate management at a later stage.

## Ethics and dissemination

This study was approved by the Ethics Committee of the University of South China (number 2022-USC-HL-222) in November 2022. Before commencing the investigation, the study participants will be informed of the objectives and procedures of the study. Specifically, we will provide a comprehensive explanation of the study to all patients, highlighting the utilization of their medical records in our research and the requirement for written informed consent to ensure their voluntary participation in the study and informed understanding. In order to guarantee complete anonymization, numerical identifiers will be utilized instead of names for all data. Research outcomes will be formally disseminated through peer review and conference proceedings.

## Study status

This study is planned to begin on March 1, 2023, and finish on January 31, 2024. Quantitative and qualitative data collection is set to be completed between March 1, 2023, and December 31, 2023. We have collected the T1 and T2 data, and we are in the process of collecting the T3 data.

## Discussion

Despite the wonderful vision of the Health China Strategy, there are still many challenges in managing chronic diseases. Patients with CHD tend to lack precise symptom management after stenting, which may increase the risk of recurrent adverse cardiovascular events, notwithstanding the huge amount of human resources and medical resources spent. This study will use a mixed-methods design to explore the symptom patterns of patients in various domains after stent implantation and subsequently clarify the management needs of patients during different time periods after surgery. Qualitative and quantitative data collection will contribute to a more comprehensive understanding of the symptom situation and management needs of patients at different times. The large sample size and three-phase survey will ensure reliable symptom patterns. The results of this study will significantly contribute to understanding individual management needs, forming the precise management of patients post-stenting, and improving the pain point of recurrent adverse cardiovascular events in patients undergoing PCI with stent implantation. Researchers can provide individualized care to patients with

different characteristics or in different postoperative periods based on the derived symptom patterns. This study is also an early attempt to use PROMIS in the cardiovascular field in China. It is meaningful to pay attention to these non-specific symptoms in patients with cardiovascular diseases, which can reflect their real feelings. In the era of "Health China," proactive health advocates for active patient participation in health management. In this study, under the concept of proactive health, patients will self-report their subjective feelings using PROMIS, helping them to proactively focus on their health problems, promote health literacy, and incorporate the active participation of patients into health management, which is a successful transition from passive medical treatment to active management. Exploring the precise management needs of patients after stenting during different time periods will contribute to the quality of proactive health oriented management and improve the health outcomes of patients.

The limitations of this study include the sample included in the quantitative phase of the study. The study populations are all from Hunan Province, China, and the increased selection bias may reduce the generalizability of our results. In addition, we did not create a complete set of management programs. Future studies could combine the different management needs during different time periods after stenting derived from this study with current relevant rehabilitation guidelines and the best practice evidence to develop a set of precise interventions and test their usability.

## Conclusions

This study is the first study to describe the dynamic patterns of symptoms presented by the transition probabilities between symptom categories and explore patient experiences across various domains after stent implantation. It utilizes a novel study design that incorporates both quantitative and qualitative methods. The study results will provide valuable insights into the development of patient health status post-stenting, as measured by PROMIS, and will provide important data for the development of early prevention of recurrent cardiovascular events. This represents a significant shift from passive medical treatment to proactive management.

## Supporting information

**S1 Checklist. This is the GRAMMS checklist.**
(DOCX)

## Acknowledgments

We gratefully thank all of the study subjects for their participation.

## Author Contributions

**Conceptualization:** Qi Wang, Yanjin Huang.

**Methodology:** Qi Wang, Chaoyue Xu, Zhiqing He, Yanjin Huang.

**Supervision:** Yanjin Huang.

**Writing – original draft:** Qi Wang.

**Writing – review & editing:** Ping Zou, Jing Yang, Yanjin Huang.

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
