## [Decision Letter · Decision Letter 0]

21 Aug 2023

PONE-D-23-13569

An exploration of proactive health oriented symptom patterns in patients receiving percutaneous coronary intervention with stent implantation: a mixed-methods study protocol

PLOS ONE

Dear Dr. Huang,

Thank you for submitting your manuscript to PLOS ONE. After careful consideration, we feel that it has merit but does not fully meet PLOS ONE’s publication criteria as it currently stands. Therefore, we invite you to submit a revised version of the manuscript that addresses the points raised during the review process.

We look forward to receiving your revised manuscript.

Kind regards,

Chiara Lazzeri

Academic Editor

PLOS ONE

Journal Requirements:

Additional Editor Comments:

The paper is well written. However some issues can be raised. Please specify the study period. We suggest to describe how sample size was calculated. Statistical analysis should be better described. We suggest to add a paragraph highlightening the novelty of the present investigation.

Reviewers' comments:

Reviewer's Responses to Questions

**Comments to the Author**

1. Does the manuscript provide a valid rationale for the proposed study, with clearly identified and justified research questions?

Reviewer #1: Yes

2. Is the protocol technically sound and planned in a manner that will lead to a meaningful outcome and allow testing the stated hypotheses?

Reviewer #1: Partly

3. Is the methodology feasible and described in sufficient detail to allow the work to be replicable?

Reviewer #1: No

4. Have the authors described where all data underlying the findings will be made available when the study is complete?

Reviewer #1: No

5. Is the manuscript presented in an intelligible fashion and written in standard English?

Reviewer #1: Yes

6. Review Comments to the Author

You may also provide optional suggestions and comments to authors that they might find helpful in planning their study.

Reviewer #1: Though the protocol is well written and used novel approach for analysis as described, there are some queries about this study protocol and are listed below.

Query 1: Line # 174: Specify the study period in the Methods session, though it’s mentioned at the end.

Query 2: Methods: Line # 222: Sample size formula looks like one used for estimation of prevalence ...Please make use of appropriate notation. What is μ2a/2 ? Is this the most appropriate method to estimate the Sample size? How do you justify the sample size for a longitudinal study based on prevalence rate of coronary heart disease admission for stent implantation?

Query 3: Line # 224: It was mentioned that the allowable error was 0.05. Is it in absolute terms or relative terms? What is the rationale for a design effect of 2 in this study?

Query 4: Line # 226: The final sample size mentioned is 430. How are you planning to recruit 430 participants from these 3 hospitals?

Query 5: Line # 246: Are the instruments including PROMIS profile-57 and others planning to use in this study, a validated ones? If so, incorporated the validation and reliability measures of those instruments.

Query 6: Line # 264: It has been mentioned that “Evon [47] and Krohe [48] concluded that 265 short-term PROMIS has good reliability and validity”. Provide more details about the measure used for describing the validity and reliability?

Query 7: Line # 271: Similar to previous query it was mentioned that “ The Chinese version of the PROMIS profile-57 has been used to assess general symptoms and 273 functions among patients with breast cancer and has been found to have good 13 reliability and validity [50]”. Provide more details.

Query 8: Line 282: Citation of the softwares should be made in standard format. Also include the software for the thematic analysis in qualitative study.

Query 9: Line 345: Provide more concrete description of Statistical analysis plan for the quantitative data. A detailed illustration of latent class analysis and latent translation analysis including the estimation procedure is necessary.

Query 10: In general, avoid typo-error that is observed in several places of this protocol.

7. PLOS authors have the option to publish the peer review history of their article (what does this mean?). If published, this will include your full peer review and any attached files.

Reviewer #1: No

---

## [Author Response · Author response to Decision Letter 0]

9 Sep 2023

Dear editor and reviewer:

Thank you for giving us an opportunity to revise our manuscript. We appreciate your positive feedback and constructive suggestions on our manuscript entitled “An exploration of proactive health oriented symptom patterns in patients undergoing percutaneous coronary intervention with stent implantation: a mixed-methods study protocol” (ID: PONE-D-23-13569). Your suggestions have allowed us to improve our paper and provide important guidance for our research. 

Taking into consideration the contributions made to our manuscript during the revision process, we have made adjustments to the order of the second and third authors, as well as added an additional author, with corresponding modifications made in the authors’ contributions section of the article.

Additional Editor Comments:

The paper is well written. However some issues can be raised. Please specify the study period. We suggest to describe how sample size was calculated. Statistical analysis should be better described. We suggest to add a paragraph highlightening the novelty of the present investigation.

RE: Thank you for your valuable suggestions.

1. In our study, the investigation time is divided into the following three periods based on cardiac rehabilitation stages in China [1]: post-stenting hospitalization, 3 months after discharge, and 6 months after discharge. We have elaborated the description of the study period in the Methods session and revised other unclear descriptions. In addition, we have renewed our study status in the manuscript. The following revisions were made: 

“The longitudinal design, which includes surveys regarding three different periods (post-stenting hospitalization, 3 months after discharge, 6 months after discharge) based on the cardiac rehabilitation stages [39]” (Line 173-176); 

“rehabilitation phase II (3 months after discharge, T2), and rehabilitation phase III (6 months after discharge, T3)” (Line 196-197); 

“at 3 and 6 months after discharge” (Line 219);

“T2: 3 months after discharge; T3: 6 months after discharge” (Line 297-298);

“We have collected T1 and T2 data, and we are in the process of collecting the T3 data.” (Line 422-423).

Reference:

[1] Committee of Cardiac Rehabilitation and Prevention of Chinese Association of Rehabilitation Medicine. [Guidelines for cardiovascular rehabilitation and secondary prevention in China 2018 simplified edition]. Zhonghua Nei Ke Za Zhi. 2018;57(11):802-810.

2. We re-checked the sample size formula used in our initial manuscript and carefully reviewed our statistical analysis. The sample size calculation formula utilized in our manuscript is applicable for studies aimed at investigating the prevalence of a specific condition which is not consistent with our study objective. Finally, we choose to specify the sample size for latent class modeling. The risk of overfitting the model to the data will increase with a higher number of subgroups relative to the sample size, and an insufficient sample size can provide difficulties with model convergence. However, there is no simple way to estimate the required sample size [2,3]. The estimation of sample size for Latent Class Model (LCM) is currently being explored. According to Finch and Bronk, it would appear that Latent Class Analysis (LCA) requires samples well into the hundreds, with most simulation studies suggesting 500 as a worthy goal in practice [4]. Latent Transition Analysis (LTA) is a statistical technique that combines cross-sectional measurement of categorical latent variables and longitudinal description of change. It is based on the method of LCA. We discussed this aspect within our team and decided to set the sample size at 500, as recommended in the literature [4]. We have deleted the text regarding sample size in the original manuscript and rephrased the description in the revised version as follows:.

“We aim to explore the symptom patterns of patients with CHD after stent implantation using Latent Transition Analysis (LTA) which is based on Latent Class Analysis (LCA) and enables the researcher to model dynamic changes. The estimation of sample size for Latent Class Model (LCM) is currently being explored. Most studies suggest 500 as the worthy goal in practice [40]. Therefore, we will recruit a total of 600 eligible patients who meet the inclusion criteria from three tertiary hospitals in Hengyang City with an expected dropout rate of 20%.” (Line 220-226).

Reference:

[2] Kongsted A, Nielsen AM. Latent Class Analysis in health research. J Physiother. 2017;63(1):55-58.

[3] Muthén L.K, Muthén B. Struct Equ Modeling. 2002;9:599–620.

[4] Finch WH, Bronk KC. Struct Equ Modeling. 2011;18:132–151.

3. We have provided a more detailed description of the statistical analysis plan for the quantitative data as follows: 

“In the quantitative study, we aim to explore the symptom categories and their dynamic shift patterns in patients undergoing stent implantation through the three-stage longitudinal survey. First, latent class analysis will be conducted separately for T1, T2, and T3 to classify post-stenting patients into different homogeneous symptom categories at each postoperative stage. A series of LCA models will be performed in order to identify the optimal classification. The number of latent classes will be gradually increased until adding an additional category no longer improves the fit index of the model. Next, we will compare the evaluation criteria of all the LCA models to determine the best model including the Akaike information criterion (AIC), Bayesian information criterion (BIC), adjusted BIC (aBIC), entropy, Lo-Mendell-Rubin likelihood-ratio test (LMR-LRT), bootstrap likelihood-ratio test (BLRT), and theoretical basis. The best model will be expected to possess lower values for the AIC, BIC, and aBIC, higher entropy, and significant LMR-LRT and BLRT results (P＜0.05) [54]. The same modeling procedure will be applied to latent transition analysis using the longitudinal data. To ensure that the latent classes can be defined consistently across time, the restriction of measurement invariance across three different time points will be implemented. The full information maximum likelihood (FIML) was performed to handle missing values in our LTA model. This method is better than most alternative missing values methods used in longitudinal studies [55]. In LTA, a patient’s latent symptom category will be allowed to change over time, and the transition probabilities to different symptom categories at different periods (T1-T2, T2-T3) will be presented, revealing the dynamic changes of individual symptoms over time. Finally, logistic regression will be used to explore factors including demographic and clinical characteristics in the general questionnaire influencing the transition of latent symptom classes at T1, T2, and T3.” (Line 364-389).

4. We have added a paragraph (Conclusions) to highlight the novelty of the present investigation. 

“This study is the first study to describe the dynamic patterns of symptoms presented by the transition probabilities between symptom categories and explore patient experiences across various domains after stent implantation. It utilizes a novel study design that incorporates both quantitative and qualitative methods. The study results will provide valuable insights into the development of patient health status post-stenting, as measured by PROMIS, and will provide important data for the development of early prevention of recurrent cardiovascular events. This represents a significant shift from passive medical treatment to proactive management.” (Line 462-469).

Reviewer #1: 

Though the protocol is well written and used novel approach for analysis as described, there are some queries about this study protocol and are listed below.

Query 1: Line # 174: Specify the study period in the Methods session, though it’s mentioned at the end.

RE: We appreciate this comment. In our study, the investigation time is divided into three periods based on cardiac rehabilitation stages in China [1]. We have defined the following the time points to provide more specific details, as follows: post-stenting hospitalization, 3 months after discharge, and 6 months after discharge. We have specifically elaborated the study period in the Methods session as you suggested and revised the related contents in other areas.

“The longitudinal design, which includes surveys regarding three different periods (post-stenting hospitalization, 3 months after discharge, 6 months after discharge) based on the cardiac rehabilitation stages” (Line 173-176);

“rehabilitation phase II (3 months after discharge, T2), and rehabilitation phase III (6 months after discharge, T3)” (Line 196-197); 

“at 3 and 6 months after discharge” (Line 219);

“T2: 3 months after discharge; T3: 6 months after discharge” (Line 297-298).

Reference:

[1] Committee of Cardiac Rehabilitation and Prevention of Chinese Association of Rehabilitation Medicine. [Guidelines for cardiovascular rehabilitation and secondary prevention in China 2018 simplified edition]. Zhonghua Nei Ke Za Zhi. 2018;57(11):802-810.

Query 2: Methods: Line # 222: Sample size formula looks like one used for estimation of prevalence ...Please make use of appropriate notation. What is μ2a/2 ? Is this the most appropriate method to estimate the Sample size? How do you justify the sample size for a longitudinal study based on prevalence rate of coronary heart disease admission for stent implantation?

RE: Thank you for raising these important questions. We re-checked the sample size formula used in our initial manuscript and carefully reviewed our statistical analysis. The sample size calculation formula utilized in our manuscript is applicable for studies aimed at investigating the prevalence of a specific condition, and μa/2 should be represented as za/2 at a confidence level of 95%. We apologize for the use of incorrect expression. Therefore, the formula isn’t the most appropriate method to estimate sample size, as you suggest, because our study aim is to explore the dynamics of symptom categories and clarify patient experiences of stenting during different time periods. 

We examined the literature on latent class analysis and latent transition class. The risk of overfitting the model to the data will increase with a higher number of subgroups relative to the sample size, and an insufficient sample size can provide difficulties with model convergence. However, there is no simple way to estimate the required sample size [2,3]. The estimation of sample size for Latent Class Model (LCM) is currently being explored. According to Finch and Bronk, it would appear that Latent Class Analysis (LCA) requires samples well into the hundreds, with most simulation studies suggesting 500 as a worthy goal in practice [4]. Latent Transition Analysis (LTA) is a statistical technique that combines cross-sectional measurement of categorical latent variables and longitudinal description of change. It is based on the method of LCA. Our team discussed this aspect and we decided to set the sample size of 500, as recommended in the literature [4]. We have deleted the original sample size and rephrased the description in our revised version of the manuscript. 

“We aim to explore the symptom patterns of patients with CHD after stent implantation using Latent Transition Analysis (LTA) which is based on Latent Class Analysis (LCA) and enables the researcher to model dynamic changes. The estimation of sample size for Latent Class Model (LCM) is currently being explored. Most studies suggest 500 as the worthy goal in practice [40]. Therefore, we will recruit a total of 600 eligible patients who meet the inclusion criteria from three tertiary hospitals in Hengyang City with an expected dropout rate of 20%.” (Line 220-226).

Reference:

[2] Kongsted A, Nielsen AM. Latent Class Analysis in health research. J Physiother. 2017;63(1):55-58.

[3] Muthén L.K, Muthén B. Struct Equ Modeling. 2002;9:599–620.

[4] Finch WH, Bronk KC. Struct Equ Modeling. 2011;18:132–151.

Query 3: Line # 224: It was mentioned that the allowable error was 0.05. Is it in absolute terms or relative terms? What is the rationale for a design effect of 2 in this study?

RE: Thank you for raising this important issue. The allowable error is in relative terms in our original manuscript. Considering the nature of the longitudinal study and the cross-sectional attribute of sample size calculation, we initially set a design effect of 2. We have changed our sample size calculation method which is now more accurate and applicable to our study aim. Some values such as the allowable error and design effect are mentioned in our manuscript. The description of the sample calculation has been revised as follows: 

“We aim to explore the symptom patterns of patients with CHD after stent implantation using Latent Transition Analysis (LTA) which is based on Latent Class Analysis (LCA) and enables the researcher to model dynamic changes. The estimation of sample size for Latent Class Model (LCM) is currently being explore. Most studies suggested 500 as the worthy goal in practice [40]. Therefore, we will recruit a total of 600 eligible patients who meet the inclusion criteria from three tertiary hospitals in Hengyang City with an expected dropout rate of 20%.” (Line 220-226).

Query 4: Line # 226: The final sample size mentioned is 430. How are you planning to recruit 430 participants from these 3 hospitals?

RE: Thank you for raising this important question. We have revised the total sample size, and the latest sample size is sufficient to explore potential heterogeneous latent classes and model their dynamic changes. We apologize for not setting out the details regarding practical arrangement for recruitment in the 3 hospitals. We have added this information in the revised manuscript as follows:

“we will recruit a total of 600 eligible patients who meet the inclusion criteria from three tertiary hospitals in Hengyang City with an expected dropout rate of 20%. Each hospital will enroll 200 patients through convenience sampling.” (Line 224-227).

Query 5: Line # 246: Are the instruments including PROMIS profile-57 and others planning to use in this study, a validated ones? If so, incorporated the validation and reliability measures of those instruments.

RE: Thank you for raising these questions. 

The Patient-reported Outcomes Measurement Information System (PROMIS) was initiated by National institutes of health (NIH) in 2004 which aims to provide a series of scientific, comparable and practical measurements to advance the concept of Patient-reported Outcomes (PROs). The PROMIS profile-57, a self-reported instrument, was developed to address the lack of generalizable and universal measures required to evaluate common symptoms and functions from patients’ perspectives [5,6]. All the informative items of the PROMIS-57 were selected based on a literature review, IRT analysis, and rounds of expert review of psychometric evaluation findings from PROMIS datasets. The original English version of the PROMIS-57 has brevity, breadth, and strong psychometric properties. 

Professor Yuan Changrong from Fudan University has been granted authorization by PROMIS Health Organization (PHO) to serve as the official representative of PROMIS in China in 2018. Her team is responsible for the translation, validation, development, and clinical promotion of the Chinese version of PROMIS. Professor Yuan’s team developed the simplified Chinese version of the PROMIS profile-57 using the Functional Assessment of Chronic Illness Therapy (FACIT) method—recommended by the PHO [7]. In accordance with the translation guideline, translations, reconciliation, back-translation, back-translation review, independent review, pre-finalization review, finalization, formatting and proofreading, harmonization, cognitive debriefing, and linguistic validation were incorporated and eventually formed the final translated version [8]. 

The translated measure was validated in participants with breast cancer from tertiary hospitals in Shanghai, China by Yuan’s team. In their study [9], the correlations between each item with its domains were all greater than 0.40 (range: 0.69–0.93, P< 0.05) and were higher than those of other domains, showing acceptable convergent validity and discriminant validity. At the same time, the correlations between PROMIS-57 item scores with the corresponding domains coefficients in the Functional Assessment of Cancer Therapy-Breast ranged from 0.32–0.56 (P<0.05), showing satisfactory construct validity. The aforementioned study also demonstrated sufficient reliability with Cronbach’s α coefficients for all PROMIS-57 domains being above the threshold of 0.70, ranging from 0.85 (fatigue) to 0.95 (physical function, anxiety).

In summary, PROMIS enables comparability of study results across different diseases and populations. It is suitable for self-reported symptom assessment in both general populations and various groups of patients with chronic disease [10]. Hence, PROMIS profile-57 is a non-specific scale and we can use the study results of Professor Yuan’s team to demonstrate its validation and reliability. We have described these aspects. The following revisions regarding the science of PROMIS and the validation of PROMIS profile-57 are:

“The Patient-reported Outcomes Measurement Information System (PROMIS) was initiated by National Institutes of Health.” (Line 245-246);

“The selection of informative items for the PROMIS profile-57 was based on a comprehensive literature review, item response theory analysis, and expert review of psychometric evaluation results from the PROMIS datasets. The original English version of the PROMIS profile-57 has excellent brevity, breadth, and robust psychometric properties. Furthermore, it has been successfully translated into over 40 cross-cultural adapted versions, all of which have shown satisfactory psychometric properties in various clinical conditions [43].” (Line 256-263);

“Prof. Yuan’s team developed a simplified Chinese version of the PROMIS profile-57 in accordance with the translation guidelines and conducted a psychometric evaluation in patients with breast cancer demonstrating acceptable convergent validity, discriminant validity, criterion validity and reliability. In their study, the correlations between each item with its domains were all greater than 0.40 in multi-trait scaling analysis, the correlations between PROMIS profile-57 item scores with the corresponding domains coefficients in the Functional Assessment of Cancer Therapy-Breast ranged from 0.32–0.56 using the Pearson correlation test, and Cronbach’s α coefficients ranged from 0.85 to 0.95 [50].” (Line 285-294).

Reference:

[5] Cella D, Riley W, Stone A, Rothrock N, Reeve B, Yount S, et al. The PatientReported Outcomes Measurement Information System (PROMIS) developed and tested its first wave of adult self-reported health outcome item banks: 2005–2008. J Clin Epidemiol. 2010;63(11):1179–94. 7. 

[6] Hays RD, Spritzer KL, Schalet BD, Cella D. PROMIS®-29 v2.0 profile physical and mental health summary scores. Qual Life Res. 2018;27(7):1885–91.

[7] Rawang P, Janwantanakul P, Correia H, Jensen MP, Kanlayanaphotporn R. Cross-cultural adaptation, reliability, and construct validity of the Thai version of the Patient-Reported Outcomes Measurement Information System-29 in individuals with chronic low back pain. Qual Life Res. 2020;29(3):793–803.

[8] Gao W, Yuan C. Translation and cultural adaptation of the Pediatric Patient-Reported Outcome Measurement Information System-Emotional Distress item banks into Chinese. J Spec Pediatr Nurs. 2021;26:e12318.

[9] Cai T, Wu F, Huang Q, et al. Validity and reliability of the Chinese version of the Patient-Reported Outcomes Measurement Information System adult profile-57 (PROMIS-57). Health Qual Life Outcomes. 2022;20(1):95.

[10] FRIES J F, BRUCE B, CELLA D. The promise of PROMIS: Using item response theory to improve assessment of patient-reported outcomes[J]. Clinical and Experimental Rheumatology, 2005, 23(39): S53-S57.

Query 6: Line # 264: It has been mentioned that “Evon [47] and Krohe [48] concluded that short-term PROMIS has good reliability and validity”. Provide more details about the measure used for describing the validity and reliability?

RE: Thank you for your valuable suggestions. We have reviewed the relevant content on the PROMIS validity and reliability tests in the aforementioned articles and have provided more detailed information in our revised manuscript. In order to ensure logical coherence in the narrative, we have moved this section to the beginning of this paragraph to illustrate the favorable validity and reliability of PROMIS. 

“Evon [41] conducted a study among patients diagnosed with chronic hepatitis C virus to evaluate the reliability and validity of 10 PROMIS short forms, including fatigue, depression, anxiety and sleep disturbance. The PROMIS measures exhibited favorable reliability (Cronbach’s α coefficient ≥0.87) and demonstrated moderate to strong associations with theoretically-similar items from specific scales assessing symptoms in patients with HCV patients (0.39-0.77). Additionally, all the data provided strong support for the structural validity of PROMIS measures using item-response-theory models. Krohe [42] performed a cognitive debriefing interview to evaluate the content validity of PROMIS short term regarding physical function, and showed that more than 90% of individuals demonstrated understanding of each item in the PROMIS.” (Line 246-256).

Query 7: Line # 271: Similar to previous query it was mentioned that “ The Chinese version of the PROMIS profile-57 has been used to assess general symptoms and functions among patients with breast cancer and has been found to have good reliability and validity [50]”. Provide more details.

RE: Thank you for this comment. We have reviewed the article authored by Prof. Yuan, the head of the PROMIS National Center (PNC-China), entitled "Validity and reliability of the Chinese version of the Patient-Reported Outcomes Measurement Information System adult profile-57 (PROMIS-57)". This study provides strong evidence supporting the validity and reliability of our measurement method. PROMIS enables comparability of study results across different diseases and populations. It is suitable for self-reported symptom assessment in both general populations and various groups of patients with chronic disease [10]. PROMIS profile-57 is a non-specific scale and we the study results of Professor Yuan’s team demonstrate its validation and reliability. We have added further details as suggested. 

“Prof. Yuan’s team developed a simplified Chinese version of the PROMIS profile-57 in accordance with the translation guideline and conducted a psychometric evaluation in patients with breast cancer demonstrating acceptable convergent validity, discriminant validity, criterion validity and reliability. In their study, the correlations between each item with its domains were all greater than 0.40 in multi-trait scaling analysis, the correlations between PROMIS profile-57 item scores with the corresponding domains coefficients in the Functional Assessment of Cancer Therapy-Breast ranged from 0.32–0.56 using the Pearson correlation test, and Cronbach’s α coefficients ranged from 0.85 to 0.95 [50]”. (Line 285-294).

[10] FRIES J F, BRUCE B, CELLA D. The promise of PROMIS: Using item response theory to improve assessment of patient-reported outcomes[J]. Clinical and Experimental Rheumatology, 2005, 23(39): S53-S57.

Query 8: Line 282: Citation of the softwares should be made in standard format. Also include the software for the thematic analysis in qualitative study.

RE: Thank you for raising this issue. We have revised the writing regarding the SPSS and Mplus software in our manuscript. We also have also included the software used in the qualitative study.

“All data will be recorded and described using IBM SPSS version 26.0 (IBMCorp., Armonk, N.Y., USA).” (Line 301-302);

“Mplus version 8.3 (Muthén & Muthén, Los Angeles, CA)” (Line 308-309);

“We will use NVivo version 10 software to manage qualitative data analysis.” (Line 347-348).

Query 9: Line 345: Provide more concrete description of Statistical analysis plan for the quantitative data. A detailed illustration of latent class analysis and latent translation analysis including the estimation procedure is necessary.

RE: Thank you for this suggestion. We have added a more detailed description of the statistical analysis plan for quantitative data, as follows: 

“In the quantitative study, we aim to explore the symptom categories and their dynamic shift patterns in patients undergoing stent implantation through the three-stage longitudinal survey. First, latent class analysis will be conducted separately for T1, T2 and T3 to classify post-stenting patients into different homogeneous symptom categories at each postoperative stage. A series of LCA models will be performed in order to identify the optimal classification. The number of latent classes will be gradually increased until adding an additional category no longer improves the fit index of the model. Next, we will compare the evaluation criteria of all the LCA models to determine the best model, including the Akaike information criterion (AIC), Bayesian information criterion (BIC), adjusted BIC (aBIC), entropy, Lo-Mendell-Rubin likelihood-ratio test (LMR-LRT), bootstrap likelihood-ratio test (BLRT), and theoretical basis. The best model will be expected to possess lower values for the AIC, BIC, and aBIC, higher entropy, and significant LMR-LRT and BLRT results (P＜0.05) [54]. The same modeling procedure will be applied to latent transition analysis using the longitudinal data. To ensure that the latent classes can be defined consistently across time, the restriction of measurement invariance across three different time points will be implemented. The full information maximum likelihood (FIML) was performed to handle missing values in our LTA model. This method is better than most alternative missing values methods used in longitudinal studies [55]. In LTA, a patient’s latent symptom category will be allowed to change over time, and the transition probabilities to different symptom categories at different periods (T1-T2, T2-T3) will be presented, revealing the dynamic changes of individual symptoms over time. Finally, the logistic regression will be used to explore factors including demographic and clinical characteristics in the general questionnaire influencing the transition of latent symptom classes at T1, T2 and T3.” (Line 364-389).

Query 10: In general, avoid typo-error that is observed in several places of this protocol.

RE: We carefully checked the manuscript revised the text accordingly to correct the typographical errors, as follows:

“implantation” (Line 75);

“January” (Line 420).

---

## [Editor Report · Decision Letter 1]

18 Sep 2023

An exploration of proactive health oriented symptom patterns in patients undergoing percutaneous coronary intervention with stent implantation: a mixed-methods study protocol

PONE-D-23-13569R1

Dear Dr. Huang,

We’re pleased to inform you that your manuscript has been judged scientifically suitable for publication and will be formally accepted for publication once it meets all outstanding technical requirements.

Kind regards,

Chiara Lazzeri

Academic Editor

PLOS ONE
---

## [Editor Report · Acceptance letter]

27 Sep 2023

PONE-D-23-13569R1 

An exploration of proactive health oriented symptom patterns in patients undergoing percutaneous coronary intervention with stent implantation: a mixed-methods study protocol 

Dear Dr. Huang:

I'm pleased to inform you that your manuscript has been deemed suitable for publication in PLOS ONE. Congratulations! Your manuscript is now with our production department. 

Kind regards, 

on behalf of

Dr. Chiara Lazzeri 

Academic Editor

PLOS ONE